# Constructing a Superior Interfacial Microstructure on Carbon Fiber for High Interfacial and Mechanical Properties of Epoxy Composites

**DOI:** 10.3390/nano12162778

**Published:** 2022-08-13

**Authors:** Ping Han, Lina Yang, Susu Zhang, Zheng Gu

**Affiliations:** 1College of Physics, Qingdao University, Qingdao 266071, China; 2Weihai Innovation Institute, Qingdao University, Weihai 264200, China; 3College of Chemistry and Chemical Engineering, Qingdao University, Qingdao 266071, China; 4Weifang Key Laboratory of Environmentally Friendly Macromolecular Flame Retardant Materials, Weifang 262715, China; 5Shandong Engineering Laboratory of Environmentally Friendly Macromolecular Flame Retardant Materials, Weifang 262715, China

**Keywords:** carbon fibers, polymer-matrix composites, hyperbranched polymer, interfacial properties, mechanical properties

## Abstract

The interface quality is crucial for the properties of carbon fiber-reinforced polymer- matrix composites (CFRPs). In order to improve the interfacial and mechanical properties of CFRPs, a superior gradient modulus interfacial microstructure is constructed on the carbon fiber (CF) surface by chemically grafting a self-assembly carboxyl-terminated hyperbranched polymer (HP-COOH). A monofilament debonding test, a short beam shear test, an impact test and a dynamic mechanical thermal analysis (DMTA) were conducted to investigate the properties of the modified composite. Prominent improvements of 79.6% for the interfacial shear strength, 51.5% for the interlaminar shear strength, and 49.2% for the impact strength, as well as superior heat-resistance properties are achieved for composites with the gradient modulus interface over those of the untreated CF composites. The mechanism for performance improvement is mainly attributed to the enhanced CF surface energy, mechanical interlocking, and chemical bonding interactions. In particular, an atomic force microscopy (AFM) test proved that the gradient modulus interfacial microstructure formed by HP–COOH could widen the interface layer thickness and buffer the sharp variations in the modulus from CF to resin, thereby transmitting an external force and reducing the stress concentration. This work provides a facile and efficient strategy for constructing a superior and versatile interface for high- performance composites.

## 1. Introduction

Carbon fiber-reinforced polymer-matrix composites (CFRPs) are ideal engineering materials that have been extensively applied in the aerospace, military and automotive fields due to their outstanding performances, such as a light weight, high strength and modulus, and prominent temperature resistance [1,2,3]. In general, the interphase between carbon fibers (CFs) and the matrix plays an important role in the performance of composites [4]. An optimum interface with strong interfacial adhesion could uniformly transmit an external load from the matrix to the reinforcement, thus avoiding an internal stress concentration in the interface region [5]. However, the interfacial interaction between CFs and the matrix is usually very poor because of the smooth and chemically inert surface of CFs [6]. Therefore, the design and construction of an appropriate interface is the key for exploiting the potential outstanding properties of CFRPs [7].

In recent years, many strategies for modifying CFs have been presented to enhance the interfacial properties of composites, including sizing, oxidation, chemical grafting, and plasma processing [8,9,10,11]. Among them, chemical grafting is known as one of the most effective modification methods due to its controllability, convenience, and energy saving, and more importantly, this method does not significantly sacrifice the strength of carbon fibers [12].

As novel polymers, hyperbranched polymers (HBPs) have been paid much attention by researchers due to their special structure and performance [13,14]. First, the abundant active functional groups in HBPs can provide many active sites to “bridge” CFs and the matrix through chemical bonds [15,16]. In addition, many polar functional groups in HBPs can be grafted onto the CF surface, which can greatly enhance the wettability and compatibility, and further promote the interaction between CFs and the matrix [17]. Furthermore, the unique dendritic structure of HBPs contains many voids and free spaces where the matrix tends to infiltrate and form a cross-linking network structure at the interface [18].

To date, many researchers have further examined the surface modification of CFs by chemically grafting HBPs to improve the interfacial and mechanical performance of CFRPs. Han et al. [19] functionalized CFs with amino-terminated polyethylenimine (PEI), where the maximal interfacial shear strength (IFSS) was enhanced by 83.6%. Shi et al. [20] chemically grafted hyperbranched polyimidazole (HBM) using nucleophilic addition polymerization, and a significant improvement of 43.4% in the impact strength was acquired. All of the above studies verify that chemically grafting HBPs could remarkably enhance the mechanical properties of composites.

Recently, a study on a hyperbranched polyester of 2,2-bis-(hydroxymethyl) propionic acid polyester-32-hydroxyl (HP–OH)-reinforced epoxy composite with a 78.2% improvement in the IFSS was reported [21]. HP–OH is a multiple hydroxy-terminated polymer prepared by two commercial monomers of 2, 2-bis-(hydroxymethyl) propionic acid and 2,2-bis (hydroxymethyl)-1,3-propanediol [22]. In addition to the above advantages of HBPs, HP–OH is more suitable for industrial production and mass application as a modifier for CFRPs due to its facile synthesis, cost-effectiveness, and environmentally friendly features.

Gangineni et al. [7] decorated CFs with graphene, graphene oxide, graphene hydroxyl and graphene carboxyl nano-fillers, and the optimal performances were obtained by the graphene carboxyl modification because the polarity and reactivity of carboxyl groups were stronger than in hydroxyl and other groups. Wang et al. [23] investigated the effect of hydroxyl and carboxyl-functionalized carbon nanotubes on the mechanical properties of composites, indicating that epoxy groups could preferentially react with carboxyl groups because the efficiency of the reaction between epoxy groups and carboxyl groups was higher than that between epoxy groups and hydroxyl groups. Based on the above research, it is speculated that the modification CFs with HBPs containing carboxyl groups would be more effective at improving the fiber surface wettability and the reaction efficiency with an epoxy matrix, as well as further enhancing the interfacial and mechanical performance of the composites. In the present work, we first employed a simple chemical technique to convert the above HP–OH polymer into a carboxyl-terminated hyperbranched polymer (HP–COOH). As a novel modifier, HP–COOH was chemically grafted onto the CF surface to modify the carbon fiber/epoxy (CF/EP) composites. To the best of our knowledge, relevant work has not been reported thus far.

## 2. Materials and Methods

### 2.1. Materials

Polyacrylonitrile-based carbon fibers (3 K, 1.76 g/cm^3^) with an average diameter of 7 μm were purchased from Sino Jilin Carbon Co., Ltd. (Jilin, China), and 2,2-Bis-(hydroxymethyl) propionic acid polyester-32-hydroxyl (molecular weight 3604) was obtained from Sigma Aldrich (Shanghai, China). The epoxy resin E-51 is composed of the diglycidyl ether of bisphenol A resin with an average epoxy value Ev = 0.51 mol 100 g^−1^, and the recommended curing cycle is conducted as follows: 2 h at 363 K, then, 2 h at 393 K, then, 3 h at 423 K. 4,4′-Methylenebis(2-ethylaniline) (H-256), a liquid aromatic amine compound, due to the advantage of convenience for formulation, low toxicity, excellent temperature and chemical resistance, is often used as the epoxy curing agent. E-51 and H-256 were provided by Shanghai Macklin Biochemical Co., Ltd. (Shanghai, China), and the structures are shown in Figure 1. Other chemicals were purchased from Sigma Aldrich and Sinopharm Chemical Reagent Co., Ltd. (Shanghai, China).

### 2.2. Methods

#### 2.2.1. Amination of Carbon Fibers

The overall grafting procedure is shown in Figure 1. Briefly, CFs were first extracted in acetone using a Soxhlet extractor at 333 K for 48 h to remove the sizing agent, which was designated as untreated CF. Subsequently, the untreated CF was immersed into a mixture of AgNO_3_ (0.01 mol/L) and K_2_S_2_O_8_ (0.1 mol/L) at 234 K for 1 h to generate CF–COOH. Afterwards, the CF–COOH was transferred into a mixture of thionyl chloride (100 mL) and N, N-dimethylformamide (DMF, 5 mL) at 351 K for 48 h to produce CF–COCl. Finally, the CF–COCl was immersed into 100 mL of ethanediamine at 353 K for 12 h to obtain CF–NH_2_.

#### 2.2.2. Synthesis of HP–COOH

The preparation of HP–COOH was conducted as follows: First, 5 g of HP–OH were placed into a three-necked flask with 80 mL of acetone. Then, 5 g of succinic anhydride were added and refluxed at 338 K for 8 h with stirring. Subsequently, the above mixture was cooled to room temperature and washed twice with ether. Finally, the obtained sediments were dried at 308 K for 12 h to produce HP–COOH.

#### 2.2.3. Chemical Grafting of HP–COOH onto Carbon Fibers

Different contents of HP-COOH (0.05, 0.1, 0.2, 0.4, 0.6 g) were mixed with 60 mL of DMF and sonicated for 30 min to produce different mixtures, respectively. 0.67 g of dicyclohexylcarbodiimide and 0.067 g of 4-dimethylaminopyridine were added into the above mixture with stirring to form a homogeneous solution. Then, the CF–NH_2_ was placed in and reacted at 373 K for 24 h. After cooling to room temperature, the CF–HP–COOH was taken out, washed with deionized water and ethanol, and dried at 338 K for 8 h. The IFSS value of different CF–HP–COOH monofilaments shows a ‘‘roof’’ shape as the HP–COOH content goes up, and the optimal content of HP–COOH is 0.2 g in this work.

### 2.3. Preparation of CF/EP Composites

The CF/EP composites were prepared using the compression molding method. Briefly, the resin matrix was composed of epoxy resin (E-51) and curing agent (H-256) at a mass ratio of 100 to 32, with stirring at 333 K for 30 min. Subsequently, CFs were thoroughly infiltrated with the matrix at a mass ratio of 68 to 32 to prepare the prepregs. Then the prepregs were put into a mold with a groove (200 mm × 6 mm × 3 mm) in the center. After that, the mold, together with the prepregs, was vacuumed, and then solidified at 363 K for 2 h, 393 K for 2 h and 423 K for 3 h, accompanied by load pressures of 5, 10 and 10 MPa, respectively. The resultant specimens were obtained after cool solidification for 12 h.

### 2.4. Characterization

The morphologies of the CF surface and the fracture surface of CF/EP composites were investigated via atomic force microscopy (AFM, NT-MDT Co., Moscow, Russia) at a scanning speed of 1 Hz, scanning electron microscopy (SEM, Dimension Fastscan, Bruker, Billerica, MA, USA) with an accelerating voltage of 3 kV and transmission electron microscopy (TEM, JEM-2100F, Japan). The relative modulus of the matrix, interface, and fibers was also investigated by AFM. Three samples were tested per technique.

A TEM specimen was prepared as follows: CF monofilaments were cut into powders and dispersed in an ethanol solution with ultrasound for 15 min. Then the above dispersion was dripped onto a microgrid copper mesh with a pipette gun and dried with an infrared lamp; this process was repeated three times. Then the copper meshes together with the fibers were dried in a drying oven until ready for testing.

Chemical compositions and structures on the CF surface were determined via Fourier transform infrared spectroscopy (FTIR, Bruker, Germany) using KBr pellets with a resolution of 4 cm^−1^, Raman spectroscopy (Renishaw-invia) with a laser excitation of 532 nm, and X-ray photoelectron spectroscopy (XPS, AXIS Supra, Kratos, Britain) with a monochromatic Al Kα X-ray source at 120 W. Three samples were tested per technique.

Thermogravimetry (TG) was measured by a thermogravimetric analyzer (N–TZSCH, Germany) under nitrogen atmosphere in the temperature range of 100 °C to 600 °C at a heating rate of 10 °C/min. Three samples were tested for the average values.

The dynamic contact angle test was conducted on a dynamic contact angle meter (DCAT21, Germany) using deionised water and diiodomethane as the testing liquids. CFs surface energy ($\gamma_{f}$), dispersion ($\gamma_{f}^{d}$) and polar component ($\gamma_{f}^{p}$) could be calculated by the following formula:(1)$$\gamma_{1}\left(1+\cos\theta \right)=2{\left(\gamma_{1}^{p}\gamma_{f}^{p}\right)}^{\frac{1}{2}}+2{\left(\gamma_{1}^{d}\gamma_{f}^{d}\right)}^{\frac{1}{2}}$$
(2)$$\gamma_{f}=\gamma_{f}^{p}+\gamma_{f}^{d}$$
where $\gamma_{1}$, $\gamma_{1}^{d}$ and $\gamma_{1}^{p}$ on behalf of test liquids (deionised water and diiodomethane) surface tension, its dispersion and polar component, respectively. Three samples were tested for the average values.

The interfacial properties of the composites were characterized by the IFSS values through a monofilament debonding test (FA620, Japan) and the interlaminar shear strength (ILSS) values through a short beam shear test (GT-7000-A2X, China) at a crosshead speed of 1 mm·min^−1^. The corresponding values can be calculated according to Equations (3) and (4), respectively.
(3)$$\mathrm{IFSS}=\frac{F_{\max}}{{\pi d}_{f}L_{e}}$$
where $F_{\max}$, $d_{f}$ and $L_{e}$ represent the maximal load (N), fiber diameter (mm) and length (mm) embedded in the epoxy droplet, respectively. The final IFSS value is the average of at least 50 valid data of three samples.
(4)$$\mathrm{ILSS}=0.75\times \frac{F_{\max}}{b\times h}$$
where $F_{\max}$ is the breaking load (N), b and h are the specimen width and thickness (mm), respectively. The final ILSS value is the average of at least ten valid data of three samples.

Impact properties could be assessed using a fall weight impact test (9250 HV, Instron, USA) at an impact velocity of 2 m·s^−1^, where the size of the specimen is 55 mm × 6 mm × 3 mm. The impact strength value is the average of at least 5 valid data points. The flexural test was operated on a universal testing machine (Gotech 7000-A2X, Taiwan, China) in a three-point loading mode with the crosshead speed of 2 mm·min^−1^ according to GB/T 1449-2005. The sample size was 20 mm × 6 mm ×3 mm and the final results were obtained from the average value of five samples.

The dynamic mechanical thermal properties were evaluated using a dynamic mechanical thermal analyzer (DMTA 8000, USA) in the three-point bending mode with a heating rate of 5.00 °C/min. The dimensions of the specimen were 50 mm × 6 mm × 3 mm and the final results were the average value of five samples.

## 3. Results and Discussions

### 3.1. Chemical Compositions of the Carbon Fibers

The chemical compositions of HP–OH and HP–COOH were verified by FTIR, as presented in Figure 2. The broader band near 3435 cm^−1^ is attributed to the stretching vibration of –OH, and the absorption peaks located at 1113 cm^−1^ and 1032 cm^−1^ correspond to the stretching and bending vibrations of C–O, respectively. Compared to HP–OH, the absorption peak at 3425 cm^−1^ completely vanishes in HP–COOH, and several new peaks appear at 1782 cm^−1^, 1415 cm^−1^, 1234 cm^−1^ and 935 cm^−1^, which are the characteristic absorption peaks of carboxyl groups. In addition, several small peaks occur between 2500 cm^−1^ and 2700 cm^−1^, which are induced by the stretching vibration and deformation vibration of O–H in –COOH [24]. The above results indicate that most of the hydroxyl groups in HP–OH have been converted into carboxyl groups after a series of changes.

The microstructure changes of various CF surfaces were analyzed by Raman spectra, as presented in Figure 3a–c. For carbonaceous materials, Raman spectra display three characteristic bands at approximately 1330–1350 cm^−1^, 1580–1600 cm^−1^ and 1500–1550 cm^−1^, corresponding to the defect and disordered structure (D band), the ordered graphitic structure (G band) and the amorphous carbonaceous structure (A band), respectively [25]. Meanwhile, the intensity ratio of the D band to the G band (I_D_/I_G_) and the A band to the G band (I_A_/I_G_) signifies the structure defects and amorphous extent with active functional groups and some heteroatoms, respectively [26].

The Raman spectra data for different CFs are shown in Table 1. The I_D_/I_G_ value shows an upward trend, and CF–HP–COOH possesses a maximal I_D_/I_G_ value of 1.13, which indicates that grafting HP–COOH transforms the CF surface from graphite crystallites to a disordered structure and provides more mechanical meshing points, thereby efficiently boosting the interlocking interaction. The I_A_/I_G_ value also presents a growing trend, which is mainly due to the growth of functional groups containing nitrogen or oxygen elements. Especially for CF–HP–COOH, many terminal carboxyl groups could not only greatly enhance the CF surface polarity, but also further react with the resin matrix and hardener to generate strong chemical bonding in the composites.

XPS was employed to detect the element and functional group compositions on the CF surface, with the results shown in Table 2 and Figure 3d–f. For the untreated CF, the C, O and N contents are 92.31%, 5.26% and 2.43%, respectively. After oxidation and amination modification, the C content drops to 69.63%, while the O and N contents distinctly increase to 12.53% and 17.84%, respectively. For CF–HP–COOH, the C content (71.10%) changes slightly, whereas the O content grows significantly to 25.31% and the N content drops to 3.59%. The main reason for these changes is the reaction of carboxyl groups in HP–COOH with amino groups in CF–NH_2_.

C1s high-resolution spectra could present the type and content of active groups, as shown in Figure 3d. The C1s spectra of untreated CF could be divided into three peaks: peak C1s (1) (284.6 eV), attributed to Csp^2^ and Csp^3^, peak C1s (2) (285.6 eV), due to C–C bonds of amorphous carbon, and C1s (3) (286.3 eV), corresponding to C–OH [27]. In the case of CF–NH_2_ (Figure 3e), the C1s spectrum presents two new peaks at 288.5 and 285.3 eV, which are attributed to –N–C=O and –C–N, respectively [28], indicating that the amination modification of CFs has been accomplished. For CF–HP–COOH (Figure 3f), the -N–C=O content dramatically increases from 2.41% of the CF–NH_2_ to 6.84%, which further proves the amide reaction between HP–COOH and CF–NH_2_. The XPS results verify that HP–COOH has been successfully grafted onto the CF surface, which agrees well with the FTIR and Raman results.

TGA measurements were taken to obtain the quantitative analysis of different carbon fibers by measuring the weight loss of fibers between 100 °C and 600 °C under nitrogen (Figure 4). The weight loss of untreated CF and CF–HP–COOH are 3.0% and 37.4%, respectively, indicating the amount of HP–COOH grafted onto the CF surface is 34.4 wt%. CF–HP–COOH exhibits excellent compatibility with the epoxy matrix due to the high grafting rate of HP–COOH [29,30].

### 3.2. Surface Morphology of Carbon Fibers

The surface of untreated CF is smooth and clean with a Ra value of 21.9 nm (Figure 5a). In addition, it can be clearly seen that there are some shallow grooves parallel to the longitudinal direction of the CF (Figure 5d,g). In the case of CF–NH_2_, the Ra value increased slightly to 28.5 nm (Figure 5b) since amino modification didn’t significantly change the surface morphology of the fibers. This could also be verified by the smooth surface of C–-NH_2_, as presented in Figure 5e,h.

In contrast, the Ra value of C–-HP-COOH dramatically soared to 79.2 nm (Figure 5c), which means that there was a greater contact area and greater friction points between fibers and the epoxy to improve the mechanical interlocking interaction at the interface. Moreover, it could be clearly observed that CF surfaces are tightly adsorbed by a transparent layer with a maximal thickness of 72.47 nm, which are the HP-COOH dendrimers grafted onto the CF surface through chemical bonding (Figure 5f,i). The HP-COOH layer could then be inserted into the matrix as a transitional region to “bridge” the reinforcement and matrix during the curing process.

### 3.3. Surface Wettability of Carbon Fibers

The wettability of the CF surface could be evaluated by its surface energy, which is shown in Figure 6. The contact angles between different fibers and the test fluids show an obvious downward trend. In contrast, the CF surface energy increases from 31.73 mN·m^−1^ for untreated CF to 62.62 mN·m^−1^ for CF–HP–COOH. Improvements in the polar and dispersion components can be attributed to the introduction of polar carboxyl groups and irregular microstructures, respectively. The growth in the CF surface energy could increase the wettability of the carbon fibers, improve the compatibility between fibers and the epoxy, and thereby enhance the interfacial adhesion and strengthen the properties of the composites.

### 3.4. Interfacial Properties of the CF/EP Composites

The interfacial properties of the composites could be estimated by the IFSS values of the CF/EP monofilament. As shown in Figure 7a, the IFSS value exhibits a distinct increasing trend from 47.6 MPa for untreated CF/EP to 85.5 MPa for CF–HP–COOH/EP. The largest IFSS value of the CF–HP–COOH/EP monofilament might be ascribed to the following mechanism: first, many polar groups can greatly improve the CF surface wettability, thereby maximizing the molecular contact between fibers and the matrix; second, carboxyl-terminated groups in HP–COOH could react not only with amino groups in CF–NH_2_ but also with epoxy groups in the resin by an open ring reaction, thus generating forceful chemical bonding at the interface; third, the characteristic dendritic structure of HP–COOH can significantly enhance the roughness and surface area of fibers, thereby forming more mechanical interlocking points. Moreover, the three-dimensional structure of HP–COOH polymers has abundant voids and space where the flowing matrix is apt to permeate, thereby forming a cross-linking network interface with a medium modulus, which could effectively transfer an external load and avoid the stress concentrations due to the sudden changes in the modulus from the matrix to the reinforcement. Consequently, these factors contribute to the improved interfacial performance of the CF–HP–COOH/EP composites.

The variations in the interface performance could be further verified by the debonding morphologies of the CF/EP monofilament. As shown in Figure 7c, a smooth and clean surface and many shallow grooves can be seen for untreated CF/EP. Therefore, it can be deduced that the interfacial adhesion between the CF and the matrix is weak, and that the interfacial failure mode is adhesive failure. For CH–NH_2_/EP (Figure 7d), many particles adhere to the CF–NH_2_ surface. Some shallow grooves could still be seen, but many of them have been filled with epoxy granules, implying that the interfacial adhesion becomes stronger, and that the interfacial failure mode might be the combination of adhesive failure and cohesion failure. However, for CF–HP–COOH/EP (Figure 7e), the shallow grooves have been completely covered with a layer of resins, and a few massive resins stuck firmly onto the CF–HP–COOH surface after debonding, indicating that the interfacial adhesion becomes more powerful, and that the interfacial failure mode should be cohesion failure. In addition, red circles show the initial position of fibers debonding from the matrix. As shown by the red circles, the residual resins at the initial debonding position of CF–HP–COOH/EP are the most abundant among the others.

The interfacial properties of the composites could also be evaluated by the ILSS values of the molded composites. As shown in Figure 7b, the ILSS values show a similar increasing trend. The CF–HP–COOH/EP composite possesses the largest ILSS value of 76.2 MPa up to 51.5% of that of the untreated CF/EP composite.

The profile-section fracture microstructures of the composites after the ILSS test were surveyed to further attest the enhancement of the interfacial properties, and the yellow arrows indicate the exposed fibers As shown in Figure 7f, the interfacial shear slips occurred in most untreated CF/EP. Moreover, the pulled-out fibers are smooth and neat, without any resin particles remaining on the untreated CF surface. Interface debonding is obvious, which indicates that the interface interaction is so poor that dragging takes place at the interface. After amino modification (Figure 7g), those fibers are partially embedded into the matrix and partially exposed, with many resin particles sticking on the CF-NH_2_ surface. These results indicate that the interfacial interaction becomes stronger, but not strong enough. Comparatively, for the CF–HP–COOH/EP composite (Figure 7h), almost all of the fibers are enveloped by a layer of epoxy with few gaps between fibers being seen. Besides, there are some large epoxy bulges protruding on the CF–HP–COOH surface. All these results again demonstrate that the interface performance of the composites has been greatly improved.

### 3.5. Mechanical Properties of the Composites

The enhancement in the mechanical performance could be verified by the cross-section fracture morphology of the composites after the flexural test, as shown in Figure 8d–f. For the untreated CF/EP (Figure 8d), fibers with a certain length were dragged out, and the yellow arrow indicates abundant holes were left due to the poor interfacial adhesion. After amino modification, the amount and length of the fibers pulled out decreased significantly; nonetheless, the interface debonding and a handful of holes still exist, but the yellow arrow indicates there are many cracks between the fibers and the matrix (Figure 8e). Comparatively, for the CF–HP–COOH/EP, neither fibers pulled out nor holes remained, the fibers and the matrix were integrated together tightly, and the yellow arrow indicates the cracks between the two disappeared (Figure 8f). These results implied that the mechanical performance of the CF–HP–COOH/EP composites was remarkably enhanced duo to the introduction of HP–COOH in the interface region.

To further explore the performance enhancement mechanism of the composites, a schematic diagram is presented in Figure 8h. For the untreated CF/EP composites, without an appropriate interphase, once subjected to an external impact, the initial cracks will perpendicularly propagate to the CF surface from the matrix due to the lack of a resistance or a buffering layer between fibers and the matrix, giving rise to the fracture of the composites under a very small force. In contrast, for the CF–HP–COOH/EP composites, HP–COOH on the CF surface could form a shielding interface layer with an intermediate transition modulus, which could deflect the crack propagation paths, thereby preventing the crack tips from directly transmitting to the CF surface. In addition, the cross-locking network structure formed by HP–COOH polymers with the CF–NH_2_ and the epoxy matrix could induce more micro-cracks when the initial crack passes through, which consumes a greater amount of energy to damage the composite. As a result, the overall load-bearing capacity of the CF–HP–COOH/EP composites rises greatly, and the mechanical performance improves remarkably.

### 3.6. Dynamic Mechanical Thermal Properties of the Composites

The dynamic thermomechanical properties of the composites are characterized by the DMTA technique. For CFRPs, the variations in the E′ value could be used to test the extent of fiber modifications and matrix–fiber interactions [31]. As shown in Figure 9a, the E′ values of the CF–HP–COOH/EP composite are far higher than those of the untreated CF/EP composite over the entire temperature range. Specifically, the E′ values increase from 67 GPa (untreated CF/EP) to 75 GPa (CF–HP–COOH/EP) under Tg and from 10 GPa (untreated CF/EP) to 17 GPa (CF–HP–COOH/EP) above Tg, with enhancements of 12% and 70%, respectively. The higher E′ value is attributed to the viscoelastic deformation of HP–COOH polymers at the interface, which results in more energy dissipation to destroy the composites. In addition, the changes in the E′ values demonstrate that the composites reinforced with CF–HP–COOH could maintain a greater hardness in the high temperature range near Tg. As explained in the previous sections, HP–COOH polymers will form a shielding layer between fibers and the matrix, which could enlarge the interface volume and thickness, as well as enhance the cross-linking density, thereby significantly inhibiting the molecular chain movement of polymers at the interface, and consequently mechanically stiffening the composites [32].

Generally, Tg can be determined through the loss factor curve. The temperature at which tanδ shows its maximal value is referred to as Tg [33]. As presented in Figure 9b, Tg increases from 134 °C for the untreated CF/EP to 143 °C for the CF–HP–COOH/EP composites, indicating that the heat-resistance property of the composites has been improved. In addition, tanδ is also indicative of the fiber–matrix adhesion at the interface. The lower the tanδ value is, the better the fiber–matrix adhesion. The tanδ value of the CF-HP–COOH/EP composite is obviously lower than that of the untreated CF/EP composite, thereby confirming that fiber–matrix adhesion interactions are enhanced by grafting HP–COOH. Moreover, the loss factor curve of the CF–HP–COOH/EP is obviously wider than that of untreated CF/EP composite, indicating that the length distribution of molecular chains in the matrix is wider for CF–HP–COOH/EP composites, since HP–COOH polymers grafted onto CF surface could react further with the epoxy and hardener to form longer molecular chains.

In summary, the DMTA results indicate that grafting of HP–COOH onto the CF surface could effectively heighten the thermomechanical performance of CFRPs.

### 3.7. Interface Analysis of the Composites

The interface microstructure and modulus changes could be investigated via AFM in force modulation and Derjaguin–Müller–Toporov (DMT) modulus mode, with the results shown in Figure 10. For untreated CF/EP composites (Figure 10a,b), there is a distinct boundary between the untreated CF and the matrix, and no transition region can be seen. Corresponding to the yellow arrow, the modulus shows a sharp drop from the untreated CF to the matrix (Figure 10c). In contrast, for CF–HP–COOH/EP composites, the boundary between the CF–HP–COOH and the matrix becomes blurred, and a transition region appears instead, which is considered to be an interface layer (Figure 10d,e). In addition, the corresponding modulus presents a gradient downward tendency from the CF–HP–COOH to the matrix (Figure 10f). The intermediate layer with a transitional gradient modulus could notably relieve the internal stress concentration and effectively transmit the external load from the matrix to the fibers, thus ultimately improving the overall performances of the composites.

## 4. Conclusions

This work reported on a facile and efficient strategy for constructing an optimal gradient modulus interfacial microstructure on the carbon fiber surface to enhance the performance of the reinforced composites. The gradient modulus interfacial layer formed by the HP–COOH polymers can effectively avoid stress concentrations caused by the sharp drop in modulus from the CFs to the matrix, thereby uniformly transmitting an external load. Meanwhile, as a transitional intermediate interface layer with the cross-linking network structure, it can also induce more micro-cracks and repeatedly deflect crack propagation. Together with the plastic deformation of hyperbranched polymers, more energy would be dissipated during the destruction of the composites. Therefore, the interfacial, mechanical, and thermomechanical performances of the composites are greatly improved. Compared to untreated CF/EP composites, the IFSS, ILSS and impact strength of CF–HP–COOH/EP composites improved from 47.6 to 85.5 MPa, 50.3 to 76.2 MPa and 43.9 to 65.8 kJ·m^−2^, with impressive increases of 79.6%, 51.5% and 49.9%, respectively. This work provided a promising approach for designing and manufacturing advanced CFRP structural materials with a desired high performance. This section is not mandatory but can be added to the manuscript if the discussion is unusually long or complex.

## Figures and Tables

**Figure 1 nanomaterials-12-02778-f001:**
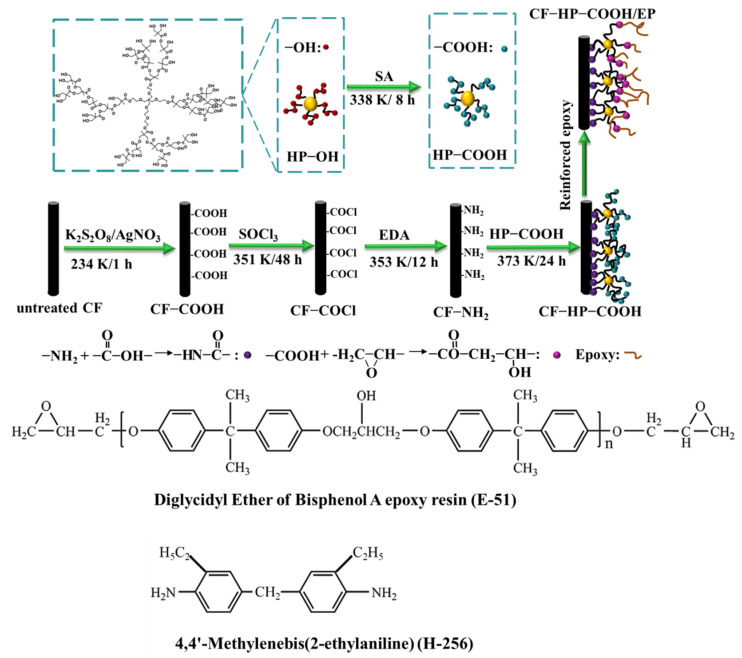
Schematic illustration of grafting HP-COOH dendrimers onto CFs surface, the main chemical interactions and the structures of E-51 and H-256.

**Figure 2 nanomaterials-12-02778-f002:**
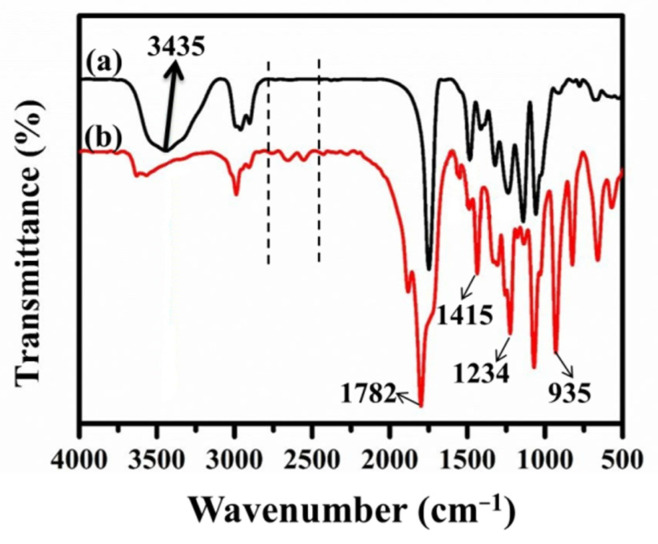
FTIR spectrum for (a) HP–OH and (b) HP–COOH.

**Figure 3 nanomaterials-12-02778-f003:**
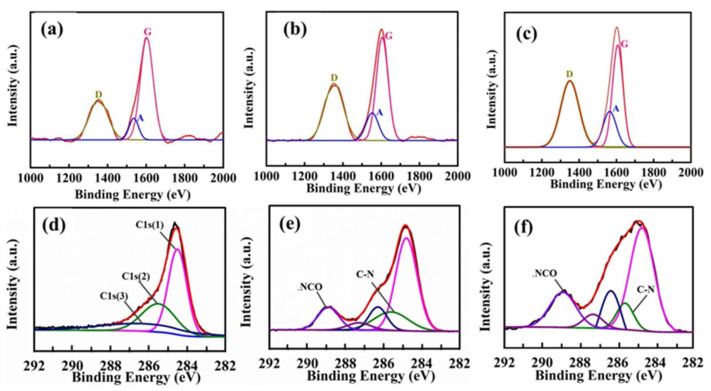
Raman spectra and C1 s high-resolution XPS spectrum for (**a**,**d**) untreated CF; (**b**,**e**) CF–NH_2_; and (**c**,**f**) CF–HP–COOH.

**Figure 4 nanomaterials-12-02778-f004:**
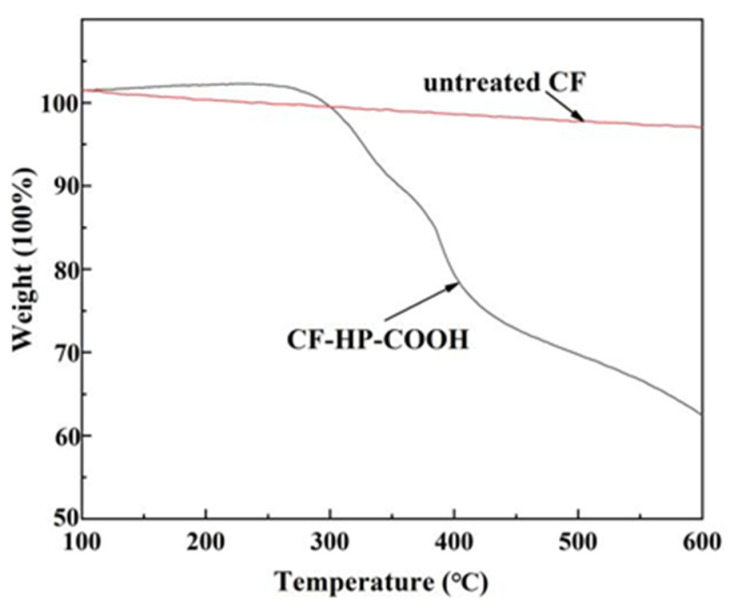
TGA curves of untreated CF and CF–HP–COOH.

**Figure 5 nanomaterials-12-02778-f005:**
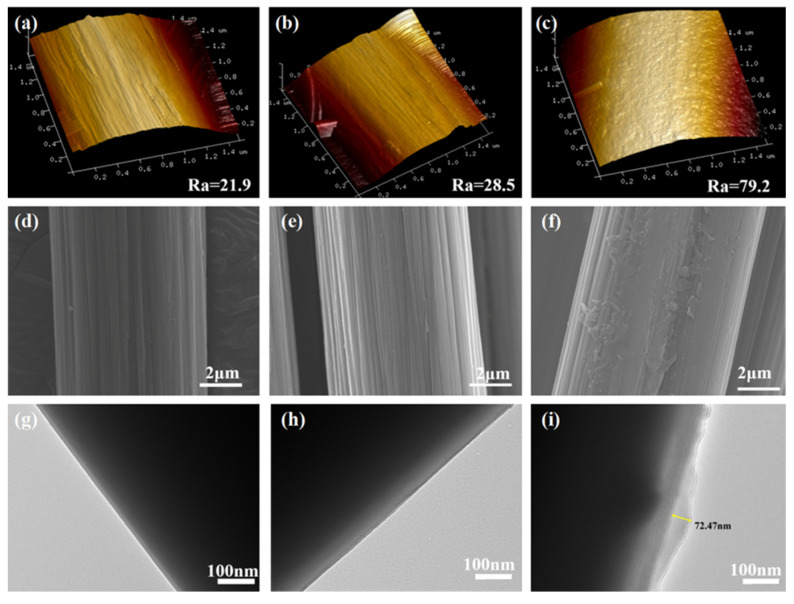
AFM, SEM and TEM images of surface topographies for (**a**,**d**,**g**) untreated CF; (**b**,**e**,**h**) CF-NH_2_; and (**c**,**f**,**i**) CF–HP–COOH.

**Figure 6 nanomaterials-12-02778-f006:**
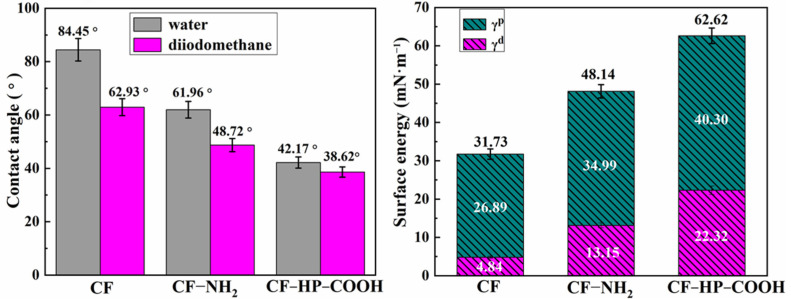
Contact angle and surface energy of different carbon fibers.

**Figure 7 nanomaterials-12-02778-f007:**
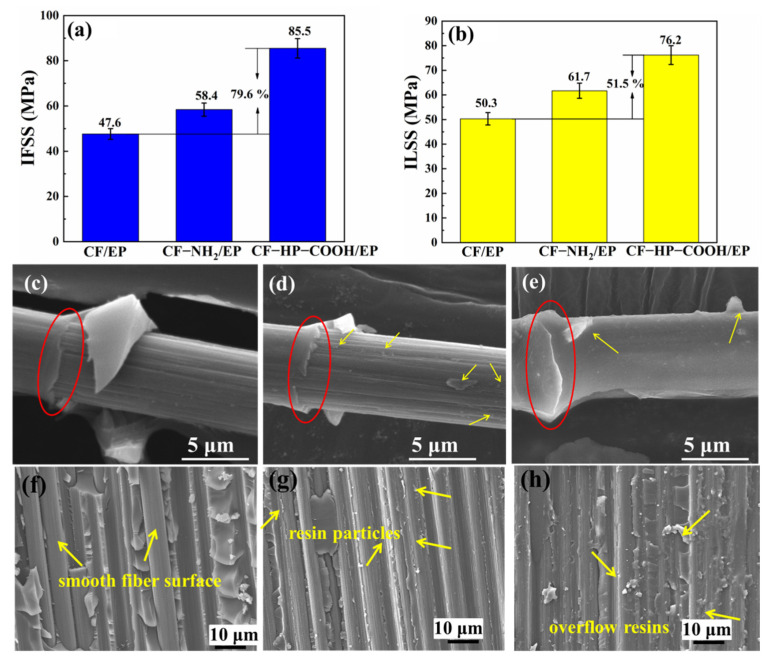
(**a**) IFSS values; (**b**) ILSS values; SEM images of debonding morphologies of monofilaments and profile-section fracture morphologies of composites for: (**c**,**f**) untreated CF/EP; (**d**,**g**) CF-NH_2_/EP; and (**e**,**h**) CF–HP–COOH/EP.

**Figure 8 nanomaterials-12-02778-f008:**
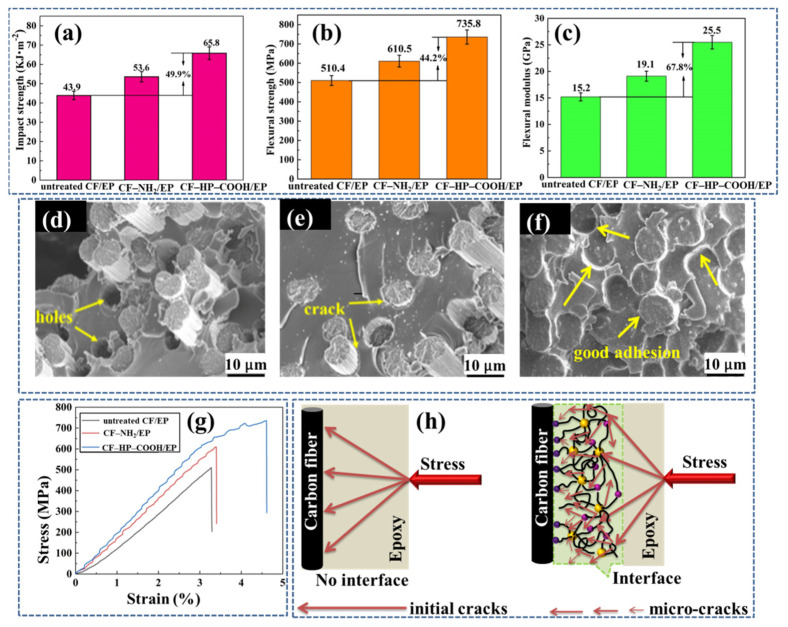
(**a**) Impact strength; (**b**) flexural strength; (**c**) flexural modulus; SEM images of cross-section fracture morphology for: (**d**) untreated CF/EP; (**e**) CF–NH_2_/EP; and (**f**) CF–HP–COOH/EP composites; (**g**) Stress-strain curves; (**h**) Schematic diagram of the properties’ enhancement.

**Figure 9 nanomaterials-12-02778-f009:**
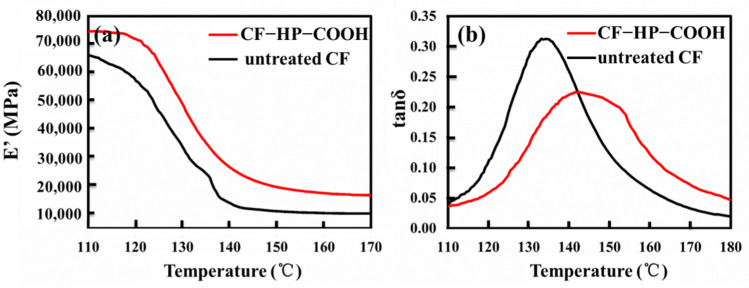
(**a**) Storage modulus (E′); (**b**) tan δ of different composites.

**Figure 10 nanomaterials-12-02778-f010:**
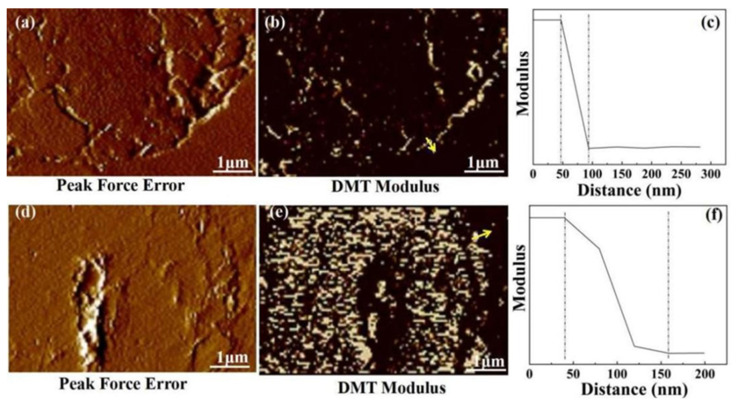
AFM images of interface region and modulus variations (along the yellow arrow) for: (**a**–**c**) untreated CF/EP and (**d**–**f**) CF-HP-COOH/EP composites.

**Table 1 nanomaterials-12-02778-t001:** Raman data of different fibers.

Samples	D	G	A	I_D_/I_G_	I_A_/I_G_
	W (cm^−1^)	W (cm^−1^)	W (cm^−1^)		
Untreated CF	1353	1600	1536	0.56	0.16
CF–NH_2_	1353	1600	1550	0.87	0.28
CF–HP–COOH	1353	1600	1550	1.13	0.44

**Table 2 nanomaterials-12-02778-t002:** Elemental compositions on the fiber surface.

Samples	Element Content (%)				
	C	O	N	O/C	N/C
Untreated CF	92.31	5.26	2.43	5.7	2.63
CF–NH_2_	69.63	12.53	17.84	18.00	25.62
CF–HP–COOH	71.10	25.31	3.59	35.60	5.05

## Data Availability

The data presented in this study are available on request from the corresponding author.

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
