# Peer review of "Constructing a Superior Interfacial Microstructure on Carbon Fiber for High Interfacial and Mechanical Properties of Epoxy Composites"

_nanomaterials, 2022, doi:10.3390/nano12162778_

Round 1

Reviewer 1 Report

The subject paper is interesting and its purpose complies with the journal’s aim and scope.

It assess the interfacial microstructure on carbon fiber for high interfacial and mechanical properties of epoxy composites.The results presented are original.

Lastly, in terms of language the manuscript is well written but a number of language and grammar corrections are necessary. 

In greater detail, the following must be improved:

Abstract: 

Please mention the characterization techniques by which you derived your results.

Introduction: 

Line 45: define performance: mechanical performance?

Line 47: line usage: “..has attracted attention”

Line 80: “… a better effect on the performance of composites.” Please explain what type of effect you are expecting to be improved.

2.4 characterization:

Please mention the number of samples you tested per technique.

Figure 4: why do you add two references in the legend of the figure? Isn’t this a result produced for the present manuscript?

Author Response

Point 1: Abstract: Please mention the characterization techniques by which you derived your results.

Response 1: We have supplemented the characterization techniques by which we derived our results in the Abstract.

Point 2:

Introduction:

Line 45: define performance: mechanical performance?

Line 47: line usage: “..has attracted attention”

Line 80: “… a better effect on the performance of composites.” Please explain what type of effect you are expecting to be improved.

Response 2:

Introduction:

Line 45:

We have replaced “performance” with “strength”;

Line 47:

 “has attracted attention” has been revised as “have been paid many attentions by researches”

Line 80:

According to aforesaid analysis, we expect a better effect on improving the fiber surface wettability and the reaction efficiency with epoxy matrix, as well as further enhancing the interfacial and mechanical performance of the composites.

Point 3:

2.4 characterization:

Please mention the number of samples you tested per technique.

Response 3:

We have supplemented the number of samples we tested per technique.

Point 4:

Figure 4: why do you add two references in the legend of the figure? Isn’t this a result produced for the present manuscript?

Response 4: Figure 4 is a result produced by the present manuscript. We cited these two references in the supplementary information of last submission, and forgot to modify. We have delete the references in the legend of Figure 4.

Reviewer 2 Report

The manuscript presents interesting findings on the surface modification of carbon fiber by chemical grafting of a self-synthesised carboxyl-terminated hyperbranched polymer. The investigations conducted by the authors are technically and scientifically interesting. The manuscript is written in clear style, but some points require additional information.

·       In my opinion, the title of the manuscript should be simplified.

·       The main characteristics of the matrix (epoxy resin) should be provided.

·       It would be useful to know the chemical nature of the hardener H-256.

·       Line 126: The statement “resin matrix was mixed with E-51 and H-256” is incomprehensible. What is the difference between the resin matrix and E-51?

·       The abbreviations ILSS (line 160) and DMT (line 393) should be clarified.

Author Response

Point 1: In my opinion, the title of the manuscript should be simplified.

Response 1: We have simplified the title as “Constructing a superior interfacial microstructure on carbon fiber for high interfacial and mechanical properties of epoxy composites”.

Point 2: The main characteristics of the matrix (epoxy resin) should be provided.

Response 2: We provided the main characteristics of epoxy resin (E-51) in 2.1. Materials, and the chemical structure in Figure 1.

Point 3: It would be useful to know the chemical nature of the hardener H-256.

Response 3: We provided the main chemical nature of the hardener H-256 in 2.1. Materials, and the chemical structure in Figure 1.

Point 4: Line 126: The statement “resin matrix was mixed with E-51 and H-256” is incomprehensible. What is the difference between the resin matrix and E-51?

Response 4: The resin matrix was composed of epoxy resin (E-51) and curing agent (H-256) at a mass ratio of 100 to 32. The statement “resin matrix was mixed with E-51 and H-256” is incomprehensible, and we have revised it.

Point 5: The abbreviations ILSS (line 160) and DMT (line 393) should be clarified.

Response 5: We have clarified the abbreviations ILSS (the interlaminar shear strength) and DMT (Derjaguin Muler Toporov).